# Development and Finite Element Analysis of a Novel Bent Bone Plate

**Joyceline Kurniawan, Shen-Yung Lin * and Wen-Teng Wang**

Department of Mechanical and Computer-Aided Engineering, National Formosa University, Yunlin 632, Taiwan
* Correspondence: sylin@nfu.edu.tw; Tel.: +886-919-051-127

**Abstract:** The main purpose of this paper is to develop a new bone plate implant design with middle bending. The bone plate design is carried out using CAD software and then tested using FEA, where output data are combined and analyzed. The simulation outcome from COMSOL Multiphysics shows that all bone plates experienced various degrees of deformation. The best bone plate would be the newly developed plate with a 10° bending angle in the middle, in comparison with the traditional flat rectangular plate, newly flat developed plate, and other bent plate with various bending angles from material or different simulation modelling. The newly developed plate bent with a 10° bending angle in the middle has an average total displacement of 4.61 nm, average von Mises stress of 0.271 MPa, and average first principal strain of $1.77 \times 10^{-6}$, making it the best choice for clinical application compared with the other bone plates analyzed.

**Keywords:** bent bone plate; implant design development; structural mechanical analysis

## 1. Introduction

Throughout time, various issues such as quality of life arise as the number of lives increases. As the population of old people increases, the need for additional help such as biomedical implants also increase, as parts of their body experience reduced functionality. Other than the aged population, the number of traffic accidents is also increasing. The victims of such accidents often need biomedical implants to assist the healing process. Biomedical implants such bone plates are used to improve the physiological state, enabling elderly and traffic accident victims to live normally. It is predicted that, by 2030, surgical replacement procedures for hip and knee will increase by over seven times in the United States. Thus, it is necessary to conduct further research into bone plate implants [1,2].

Excessive load on a human bone surpassing the maximum load it can bear usually leads to bone fracture. Bone fracture with a fragment greater than 5 mm requires additional support such as a bone implant for recovery, owing to the disappearance of the bone's self-healing ability. A bone implant secures the pieces in place, allowing it to realign and restricting exaggerated movement, thus speeding up the restoration process. Commonly, a thin rectangular metallic plate dimension of 180 mm × 30 mm × 3 mm (length × width × thickness) is implanted onto the broken bone. The bone plate usually exists with several holes designated for screw fixation [3,4].

Sometimes, it is necessary to shape the bone plate to follow the contour of the bone. This is done to avoid any loss of reduction, as well as when the lag screw is not being utilized across the fracture. Therefore, prior to the bone plate being applied to the patient's fracture, it is bent using bending tools such as hand-held bending pliers, bending press, or bending irons [5]. Thus, the bone implant is designed with a bent feature to make the application process easier.

It is necessary to understand the average measurement of the bone that will be implanted, as human bodies differ from each other. In this research, the bone plate is designed based on the adult femoral shaft, thus understanding the average size of the femoral shaft

is important. Bone plate implants usually have holes meant for screws, used as a common fixation method. A cortical screw is commonly used for bone plate fixation onto diaphyseal bone like the femoral bone. Understanding the load experienced by the bone during any kind of activity is also necessary, especially for a load bearing bone like the femoral bone. Each human body part bears distinct force while a certain action is being performed. This load may lead to fracture if it exceeds the threshold a bone could bear. As this implant is designed for the left femoral bone, the load experienced on various components while doing particular activities. For instance, during self-selected speed walking, the medio/lateral component endure force of $-11.1 \pm 0.9$, anterior/posterior component endure force of $-23.8 \pm 1.7$, and proximal/distal component endure force of $116.1 \pm 3.0$, in a percentage of body weight. Different speed of walking also produces different force endured by the femoral bone. The medio/lateral component endure force of $-11.7 \pm 1.0$ while walking in low speed, and $-11.3 \pm 1.0$ while walking in high speed. Similarly, anterior/posterior component endure force of $-20.7 \pm 1.6$ while walking in low speed and $-24.8 \pm 1.8$ while walking in high speed. Likewise, proximal/distal component endure force of $109.3 \pm 2.6$ and $121.3 \pm 3.3$ while walking in low speed and high speed respectively. [5–7].

There are various mechanical tests that can be simulated using FEA software like COMSOL, such as von Mises stress and first principal strain, in order to observe the mechanical strength of a 3D model. Von Mises stress is commonly used in mechanical strength analysis, as it is equal to the effective stress experienced. Principal or orthogonal strain shows both minimum (third) and maximum (first) strain in the normal direction where there is no shear strain. The first (maximum) principal strain is usually a positive value, expressing the increase in either the length or thickness of the model. These two, along with total displacement, are commonly used in an FEA software simulation when the mechanical strength of a model needs to be analyzed [8–10].

In order to ensure a successful recovery, a bone plate should be able to lock the bone fragments in place, preventing unnecessary movement that might prolong the recovery time. A bone plate would be feasible if the stress experienced is below the maximum limit for plastic deformation. A titanium alloy (Ti-6Al-4V) has a maximum load limit of $1060 \times 10^6$ Pa; if this is exceeded, the implant will break. Once the force is applied, the implant should not break, in order to keep the bone fragments together. These is the most important factor during bone plate design and FEM analysis [3,11–13].

This research aims to develop a bone plate implant for the left femoral bone. The implant is designed using SOLIDWorks with several bending angles, which is believed to be stronger than the traditional flat plate or the newly developed flat plate. The bent feature is also believed to provide a better structural synchronization to the bone surface, making it easier to follow the bone contour for the bone plate fixation. Several bending angles are applied to the design to observe which bending angle would be more suitable for this particular design, and how the bending angle would affect the deformation of the bone plate. This will be investigated through finite element analysis using COMSOL Multiphysics as the software, comparing the traditional rectangular plate with the newly developed plate.

## 2. Materials and Methods

In order to predict the mechanical reliability of a design, a finite element analysis (FEA) or finite element method (FEM) is used. A numerical method is utilized during the structural study of a model or design with a specific designated material. Certain load and boundary conditions are also assigned to the model, which is then meshed and analyzed. FEA is an essential step as it avoids any unnecessary procedure, especially prior to prototype production. This will save not only any unnecessary cost, but also time, while also indicating the mechanical strength of the model. Nowadays, FEM is used because of its accuracy and precision and its ability to study complicated geometries such as bone plate implants. However, a physical mechanical test would still be necessary as FEA cannot be considered to fully replace the laboratory analysis [14–16].

### 2.1. Theoretical Aspect

Human bones vary in size. The average length of the femoral shaft from the base to the apex is 374.0 mm, with a standard deviation of $\pm$26.2 mm. Based on this knowledge, the total length of the developed implant is designed to be around 260 mm, instead of the traditional 180 mm rectangular plate. In this design, the screw hole is designed for a 4.5 mm cortical screw with a 4.5 mm thread diameter and 8 mm head diameter.

Similarly, a different action may lead to a different load or force acting on a certain body part. As explained in the introduction above, various activities produce different force endured by the femoral bone. It is believed that the activity of walking at a self-selected speed is the most representative action, as people would have their own walking paces in their daily lives. As the average weight from 20 volunteers is 627 $\pm$ 145 N, the load used for simulations will be calculated using the following formula:

$$Actual\ Load[\text{N}] = \frac{Reported\ Force[\text{BW\%}]}{100\%} \times Body\ Weight[\text{N}] \tag{1}$$

Based on research done by D'Angeli, et.al. (2013), the actual load of the patients with average weight of 627N, walking at a self-selected speed will be 727.947 N in the proximal/distal direction, $-149.226$ N in the anterior/posterior direction, and $-69.597$ N in the medio/lateral direction. These values will be included in the FEA simulation using COMSOL Multiphysics as the load applied on the screw holes [6].

### 2.2. Materials

Generally, a biomedical implant uses material with great biocompatibility and a low stress shielding effect like titanium alloy. In contrast to pure titanium, titanium alloy has an improved mechanical strength of almost twice that of pure titanium, while also being light weight and highly resistant to corrosion. Titanium alloys are commonly used in medical implants owing to their properties, which help in speeding up the healing time while also being strong enough to realign the broken bone fragments. Patients also find the application of titanium alloy as a biomaterial to be comfortable [3,17].

Therefore, titanium alloy like Ti-6Al-4V is a very reasonable choice as a biomedical implant material, whose material properties are presented in Table 1.

**Table 1.** Titanium alloy 6Al-4V mechanical properties [18,19].

| No. | Mechanical Properties | Value |
|:---:|:---:|:---:|
| 1. | Density | 4430 [kg/m$^3$] |
| 2. | Young's Modulus | 113.8 [GPa] |
| 3. | Poisson's Ratio | 0.342 |
| 4. | Thermal Expansion Coefficient | 8.6 [μm/m·°C] |
| 5. | Yield Strength | $880 \times 10^3$ [MPa] |
| 6. | Tensile Strength | $950 \times 10^3$ [MPa] |

### 2.3. Geometrical Model of a Bone Plate

Prior to performing any FEA analysis, a bone plate is designed using CAD software like SolidWorks. By utilizing the average size of the human femoral bone, a bone plate is designed with an overall length of 260 mm, width of 30 mm, and thickness of 3 mm, as can be seen in Figure 1. Here, during the design process, the newly developed flat plate (Figure 2) is bent in the middle with the bending angle varying from 10° to 50° (Figure 3), in order to find out which one is the best. These plates will be compared to the conventional bone plate with a size of 180 mm in length and the same width and thickness, as can be seen in Figure 1. After the design is complete, it is then exported into a format that could be read by the FEA software, such as COMSOL Multiphysics. The simulation is performed

using COMSOL Multiphysics 5.2, version 5.2.0.166 by COMSOL AB, Stockholm, Sweden. The simulation is done on a PC using Microsoft Windows 10 Pro 64-bit with Processor Intel® Xeon® CPU E3-1230 v3 at 3.30GHz (8 CPUs) by Intel Corporation©, California, USA; 32,768 MB RAM, and 455 GB storage.

During the FEA simulation, the back of the plate is constrained, as it will be sticking to the bone (Figure 4). The boundary load is applied on every wall of the screw hole (Figure 5) to observe the deformation when the plate is fixated using screws onto the bone. Although, generally, not all screw holes will be used, the load is still applied onto all screw holes as the deformation of all screw holes should be analyzed. Various types of tests are conducted using COMSOL Multiphysics, in order to determine the best bone plate, under the same condition. The results observed are displacement, von Mises stress, and first principal strain.

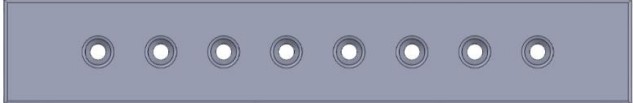

**Figure 1.** Traditional flat plate.

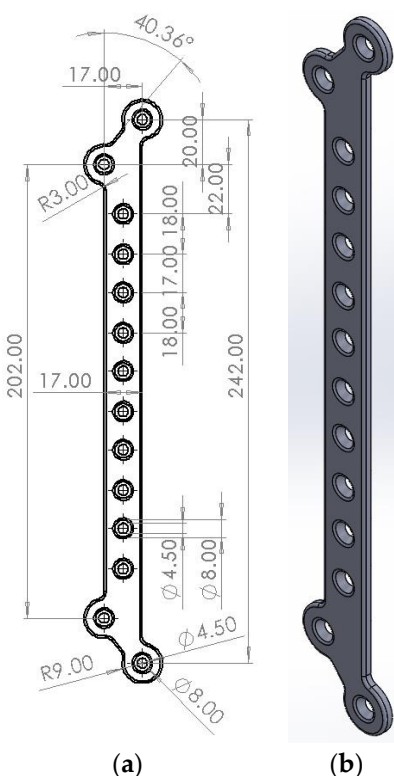

(**a**)  (**b**)

**Figure 2.** Developed plate CAD (flat); (**a**) 2D and (**b**) 3D.

### 2.3.1. Traditional Flat Plate

Traditionally, bone plates are designed to be small and flat, with no sharp edges, to hold bone fragments together. They are usually intended to be versatile and comfortable for patients, as well as to promote bone healing. Figure 1 illustrates the CAD of the traditional flat plate, with dimensions of only 180 mm in length, 30 mm in width, and 3 mm in thickness. There are eight holes integrated to provide space and support for the screw placement as the fixation method. Although it might not be necessary to use all of the holes given in the design, it is still necessary to provide and simulate these holes to see if they can actually provide stability and anchor for bone fragments lying below the bone plate. The edges on the top surface of the bone plate are also given a fillet feature to reduce any hard edges, avoiding the possibility of further injuries to the patients.

However, when a load-bearing bone like the femoral bone is broken, a bigger support is necessary.

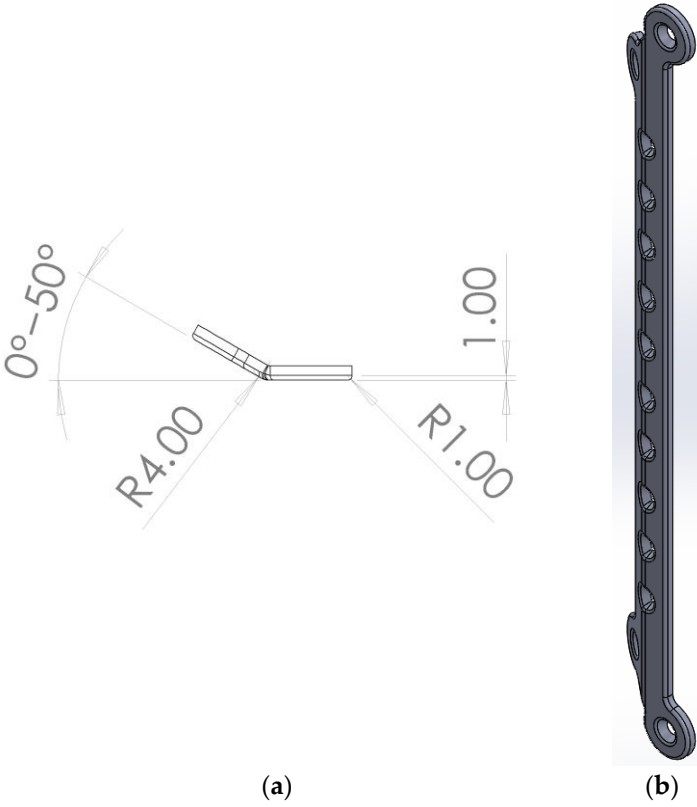

**Figure 3.** Developed plate CAD (bent); (**a**) 2D and (**b**) 3D.

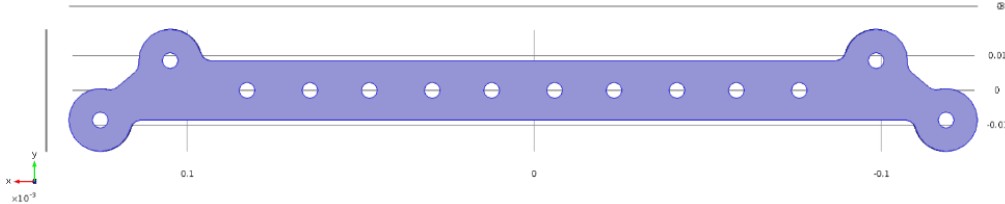

**Figure 4.** Fixed constraint on the developed plate.

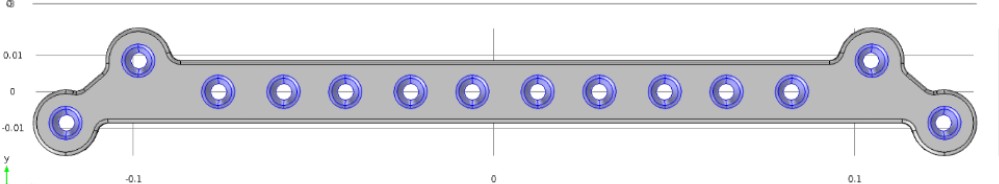

**Figure 5.** Boundary load on screw holes of the developed plate.

### 2.3.2. Developed Plate Design (Flat)

A longer bone plate is designed, with circular features given on each end to promote a better anchor as a wider area is covered, so that the healing process could occur more efficiently. Figure 2 shows the geometry of the newly developed plate before it is bent, where Figure 2a illustrates the 2D drawing along with the dimension, and Figure 2b illustrates the 3D perspective view of the newly developed bone plate. The implant is developed specifically for the human femoral bone, with a general shaft length of 374.0 mm. Therefore, the overall length of the implant can be increased to 260 mm, with the same overall width (30 mm) and thickness (3 mm).

With the same overall width as the traditional flat plate, the width on the middle shaft is reduced to only 17 mm in order to avoid any unwanted side effect caused by too much contact with a foreign object like an implant. It is also done to reduce the amount of the foreign material inside the body, aiming to avoid any side effect that might occur. This side effect may be avoided as the implant material is titanium alloy Ti-6Al-4V, which has great biocompatibility. The traditional flat bone plate is also commonly designed with several screw holes, meant for fixation to the bone. It normally has some screw holes with similar intervals, which is also implemented in the design development, with an overall diameter of around 4.5 mm, intended for cortical screws—this type and size is commonly used for long bones like the femoral bone. It may not be necessary to use all screw holes to avoid any redundant drilling into the bone. Drilling into the bone may reduce its mechanical strength, which is essential especially for a load-bearing bone like the femoral bone, hence it should be avoided. These screw holes are provided for the sake of flexibility to be used in clinical situations, as the surgeons can choose which screw hole they want to use.

### 2.3.3. Developed Design (Bent)

Traditionally, flat bone plates are available in a smaller size with a flat contour. However, as the size of the bone plate is increased in this development, a wider area of the bone is covered. While this bone plate is developed for a long bone like the femoral bone, the flat developed bone plate will not be able to cover nor protect the bone well when the edges are lifted up and not touching the bone surface. This phenomenon may occur because the flat developed bone plate is attached to the long bone, which has a cylindrical shape that is similar to a pipe. Nowadays, it is also becoming more common for patients to have an implant specifically designed for them. Therefore, the idea of the implant being bent to follow the bone contour emerged, especially implants designed for long bones like the femoral bone. This is illustrated in Figure 3, where the bending angle can be seen in the CAD design shown in Figure 3a. Figure 3b illustrates the 3D CAD to provide a better visualization on the newly developed bone plate when it is bent.

The middle curvature of the newly developed bone plate aims to follow the bone surface contour better than the flat developed plate. It is also hypothesized that it is better for the developed implant to be bent as it will not just hold the bone fragments better, but also give more comfort to patients. The bent developed plate is predicted to reduce the recovery time and be more comfortable, and thus is a better choice for clinical use.

### *2.4. FEA Boundary Condition*

Before undergoing solid mechanics analysis using COMSOL Multiphysics, a boundary condition and load are applied to the bone plate.

### 2.4.1. Fixed Support

All plates are constrained on the bottom surface, as shown in Figure 4. Figure 5 illustrates the location of the load, which is applied to all parts of the screw holes along the bone plate, simulating the force exerted by the bone screw after fixating the bone plate.

### 2.4.2. Load

Although it is not mandatory to use all of the screw holes each time, it is still necessary to find out the mechanical properties of each of the hole; therefore, all of the screw holes are simulated at the same time. This condition is assumed to be experienced by the implant when used by the patient while carrying out their daily activities.

### 3. FEA Results and Discussion

COMSOL Multiphysics is utilized as the FEA software to perform solid mechanics analysis on the bone plate. The result will be compared and analyzed between the traditional flat plate and the newly developed plate. The FEA analysis will also examine the newly developed plate, which is bent in the middle, with a bending angle ranging

between 0° (flat) up to 50°. A stationary study is applied, as only a static condition wants to be observed. The analyses performed are total displacement, von Mises Stress, and first principal strain, of which the deformation and FEA results are illustrated below in Figures 6–26. After the simulation is carried out, the data obtained through the COMSOL Multiphysics are inserted into Microsoft Excel to make graphs, in order to make it easier to analyze. The summary of all analyses is shown in Figures 27–29, with the data taken from Tables 2–4.

### 3.1. Traditional Flat Plate

Figures 6–8 show the simulation result of the traditional flat plate. The result illustrates that the traditional flat plate does not break from the load applied. The maximum von Mises stress is only 3.57 MPa, which is still below the maximum load limit of Ti-6Al-4 V of 1060 MPa. This is ideal for clinical use, as it indicates that the plate is strong enough to not breaking during activities. The maximum displacement of the traditional flat plate is only 48.8 nm, with an average of 3.63 nm. As can be seen from the graph shown in Figures 27–29, generally, the flat plate holds the top position in all analyses, except for Figures 28 and 29, illustrating that the bone plates bent at 20° and 50° have higher von Mises stress and first principal strain. This shows that the load distribution and deformation can be improved once the plate is bent to a certain angle.

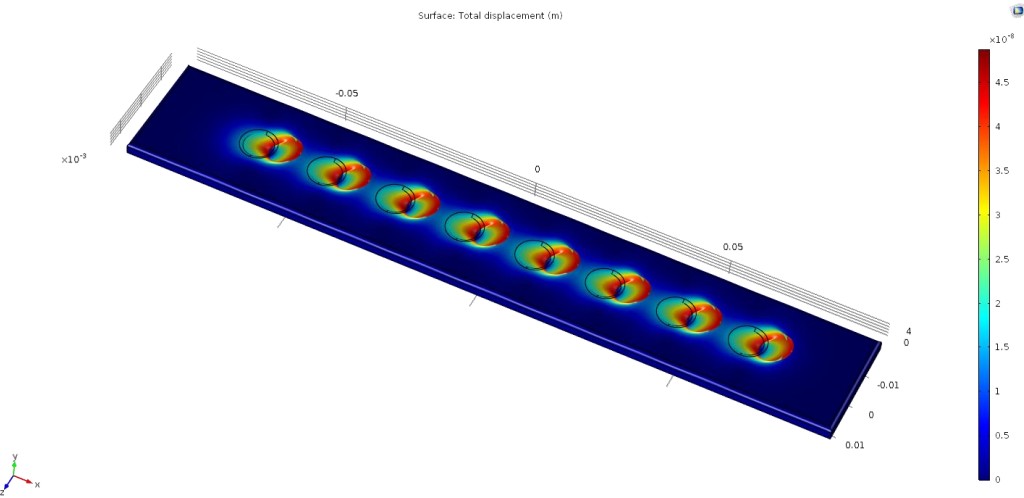

**Figure 6.** Total displacement of traditional flat plate.

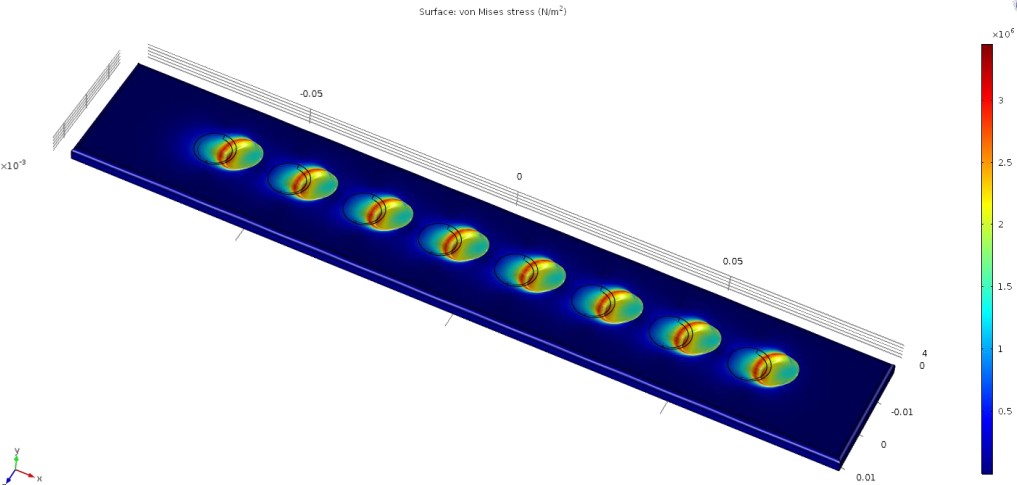

**Figure 7.** Von Mises stress of traditional flat plate.

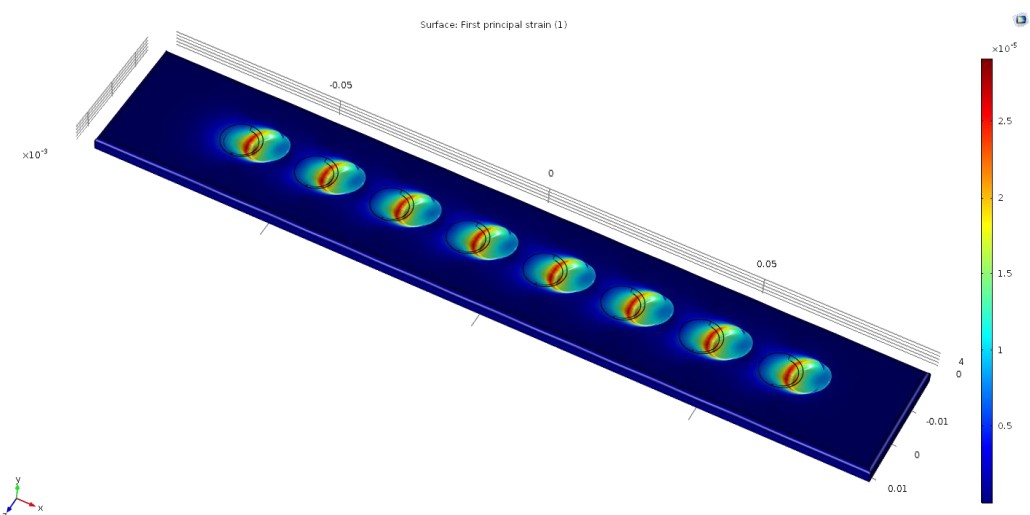

**Figure 8.** First principal strain of traditional flat plate.

### 3.2. Flat Newly Developed Plate Design

The FEA result of the newly developed flat bone plate is illustrated in Figures 9–11. Generally, this implant is stronger and more stable than the traditional flat plate. Based on the data in Table 2, the newly developed plate has a maximum displacement of only 39%, maximum von Mises stress of only 42%, and maximum first principal strain of only 44% of the traditional flat plate. It, however, has an average displacement of 4.53 nm or about 25% higher; and both average von Mises Stress and first principal strain about 18% higher, in contrast to the traditional flat plate. The length of the plate may be the cause of the improvement in the results of all analyses, considering that the increase in length helps to decrease the magnitude of the load acting per point area. The circled body on both ends of the plate also increases the total surface area, increasing the grip strength while also reducing the force.

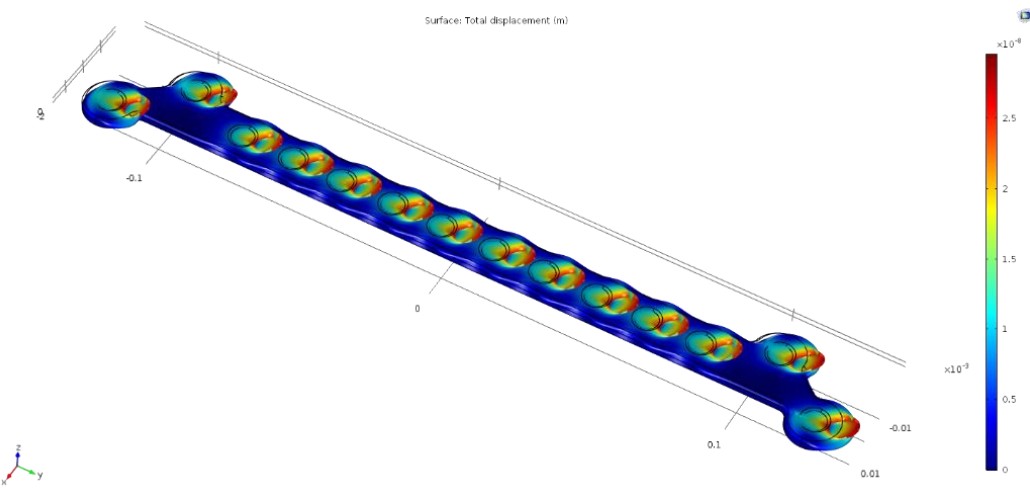

**Figure 9.** Total displacement of newly developed plate.

### 3.2.1. Bent Newly Developed Plate Design

The newly developed plate is bent in the middle, with a bending angle ranging from 0° up to 50°, with an interval of 10°. The FEA performed on the bent plate is the same as the traditional flat plate and the newly developed flat plate, the result of which is shown in Figures 12–26, illustrating the total displacement, von Mises stress, and first principal strain. It is necessary to compare the bent newly developed plate to the flat newly developed plate and the traditional flat pate in order to see if there is any improvement in the bone plate. It

is also necessary to see which one is the best from all of the simulated bone plates, so that further development could be happen to make it applicable in the clinical world.

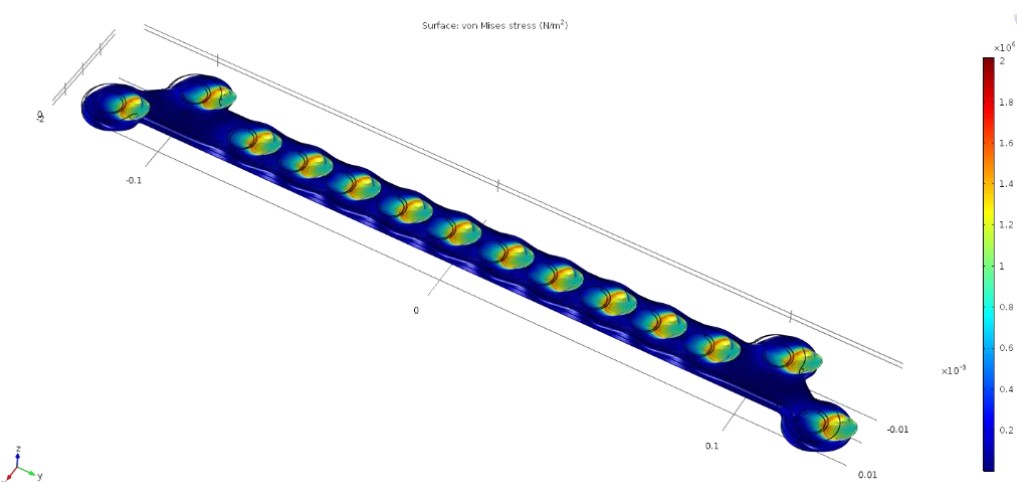

**Figure 10.** Von Mises stress of newly developed plate.

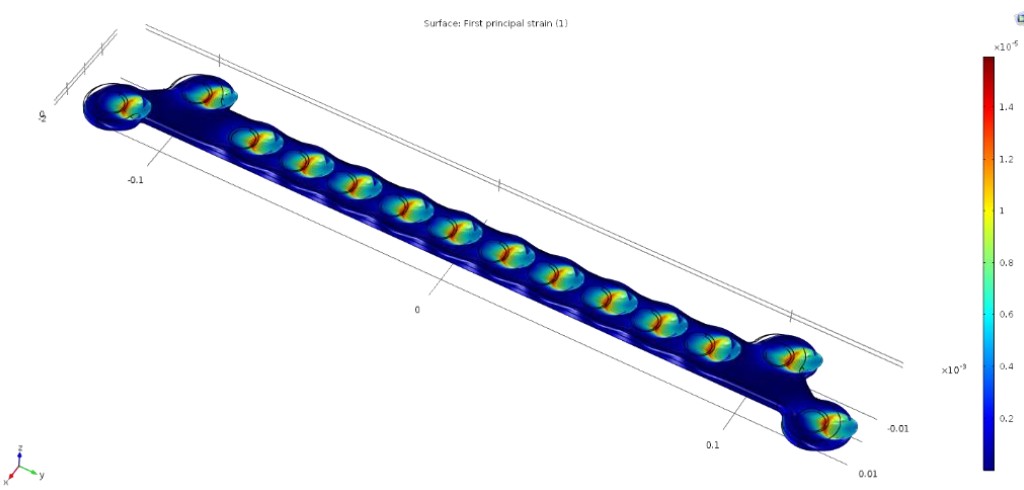

**Figure 11.** First principal strain of newly developed plate.

Bent, 10°

The simulation results of the newly developed bone plate bent with a 10° bending angle are illustrated in Figures 12–14, exhibiting the best result in contrast to other bone plates. The computational analysis shows that the maximum displacement is only 29.6 nm, same as the newly developed flat plate, with an average of 4.61 nm, as can be seen in Table 2. The maximum von Mises stress is only 19% compared with the traditional flat plate and only 46% compared with the developed flat plate. Similarly, the maximum first principal strain is only 20% of the traditional flat plate and only 44% compared with the developed flat plate. This shows a significant improvement once the developed plate is bent. However, it has average displacement, von Mises stress, and first principal strain of about 118–125% of the traditional flat plate and 106–109% of the developed flat plate. This shows that the mechanical properties of the newly developed bone plate, bent at 10°, is the most suitable compared with the other bent developed plate.

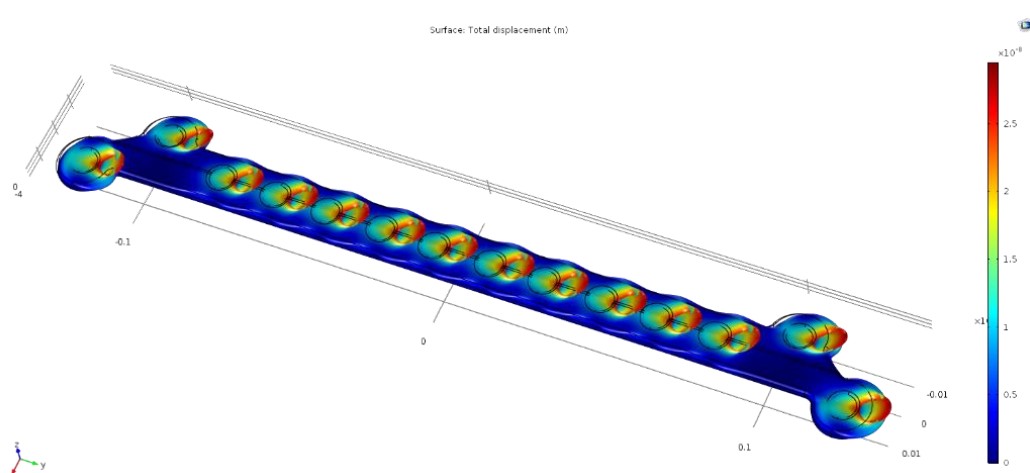

**Figure 12.** Total displacement of newly developed plate bent for 10°.

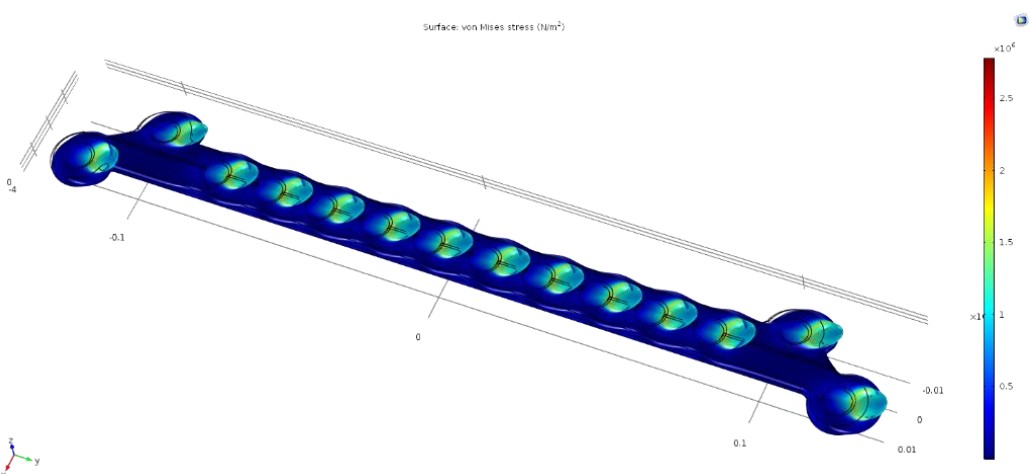

**Figure 13.** Von Mises stress of newly developed plate bent for 10°.

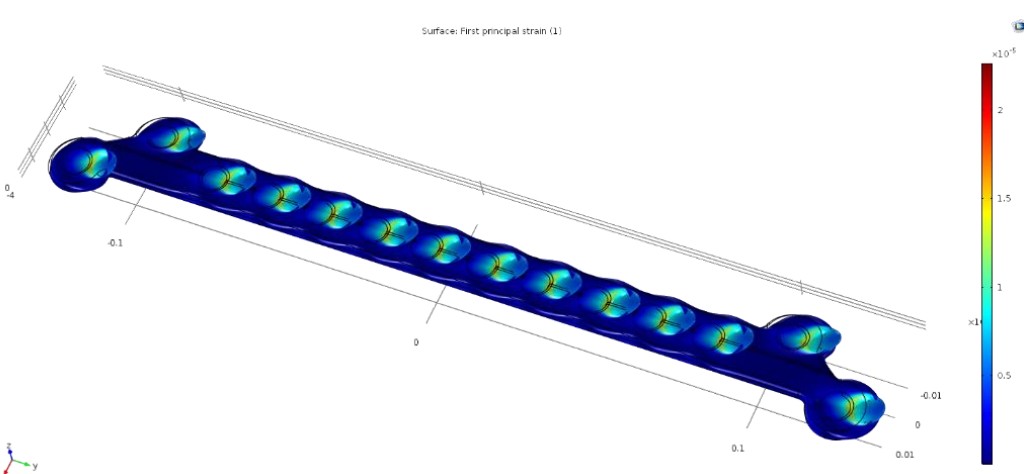

**Figure 14.** First principal strain of newly developed plate bent for 10°.

Bent, 20°

The developed bone plate bent with a 20° bending angle shows a similar result to the bone plate bent at 10°, as illustrated in Figures 15–17. When compared with the traditional flat plate, the newly developed plate bent at 20° shows a maximum displacement of about 38%, with the average almost 30% higher. However, the maximum von Mises stress and the

maximum first principal strain are 108% and 114% of the traditional flat plate, respectively, and 118% and 131% of the newly developed flat plate, respectively. Although this shows there is improvement in the displacement, both the von Mises stress and first principal strain occurring on the bone plate will make it not stable and may lead to it breaking in the future during long-term use.

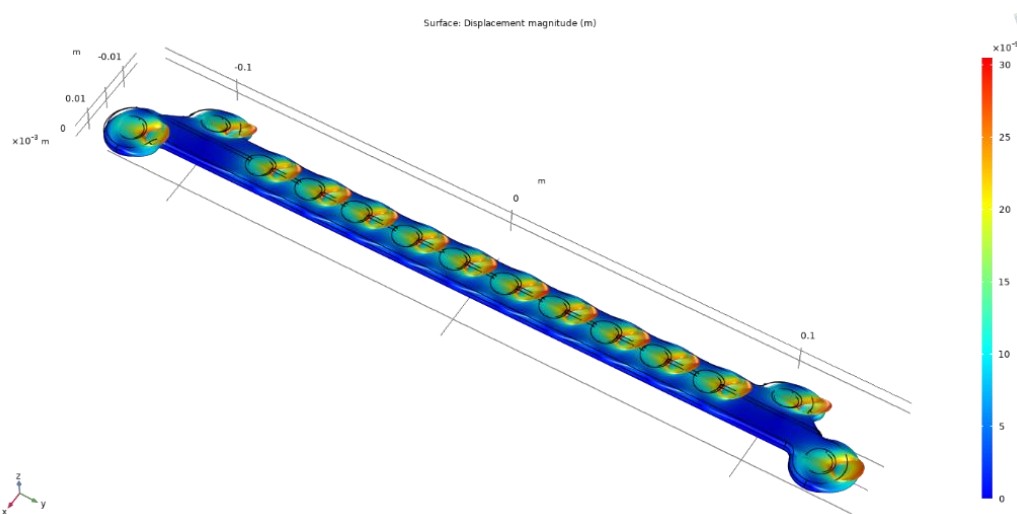

**Figure 15.** Total displacement of newly developed plate bent for $20°$.

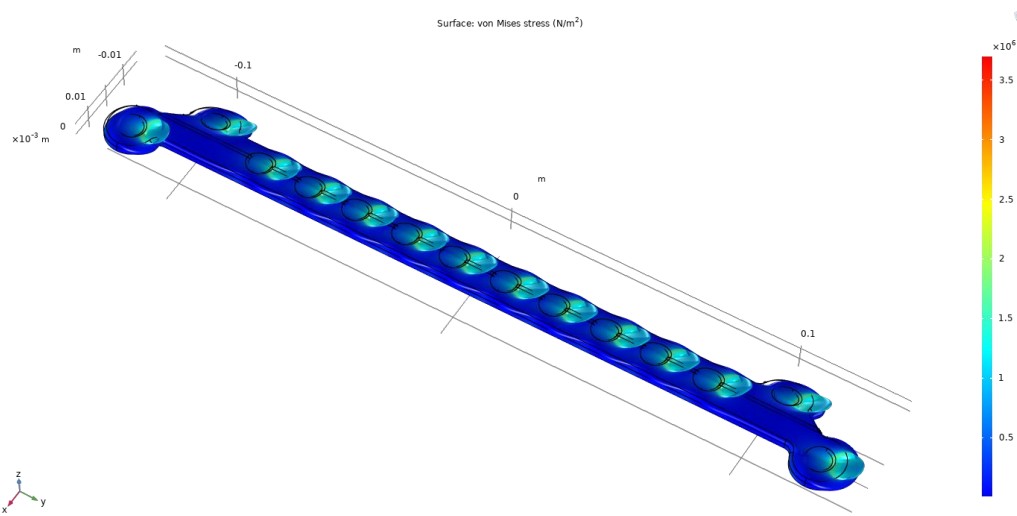

**Figure 16.** Von Mises stress of newly developed plate bent for $20°$.

Bent, $30°$

Figures 18–20 show the deformation of the newly developed plate bent at $30°$. The plate developed and bent at $30°$ shows generally the least maximum value in all aspects, but it has an average value about 20–30% higher in all aspects, in contrast to the traditional flat plate. Although this plate has the least maximum value, the high average value in all aspects will make this plate too unstable for clinical use and may cause discomfort and even breaking when the plate deforms that much. This is shown despite that the traditional flat plate is a better choice than this one.

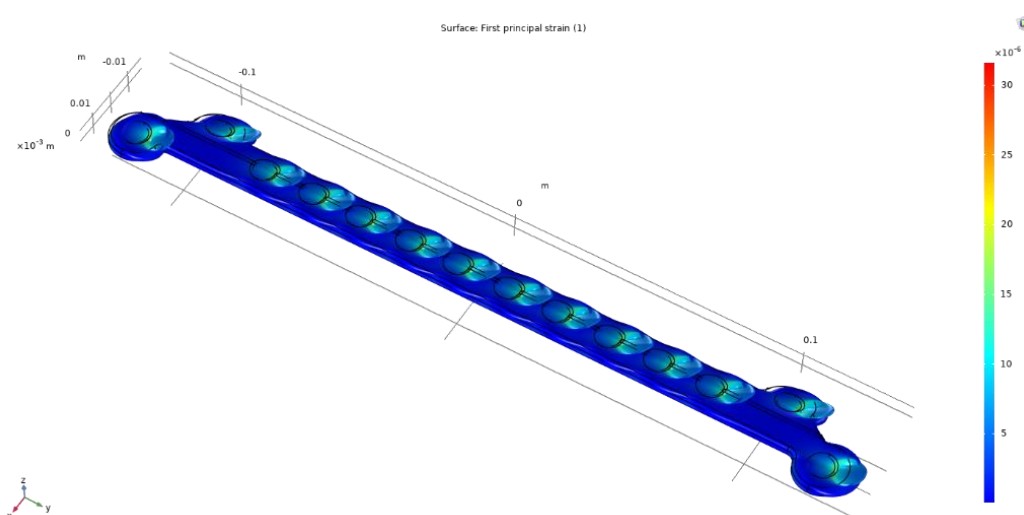

**Figure 17.** First principal strain of newly developed plate bent for 20°.

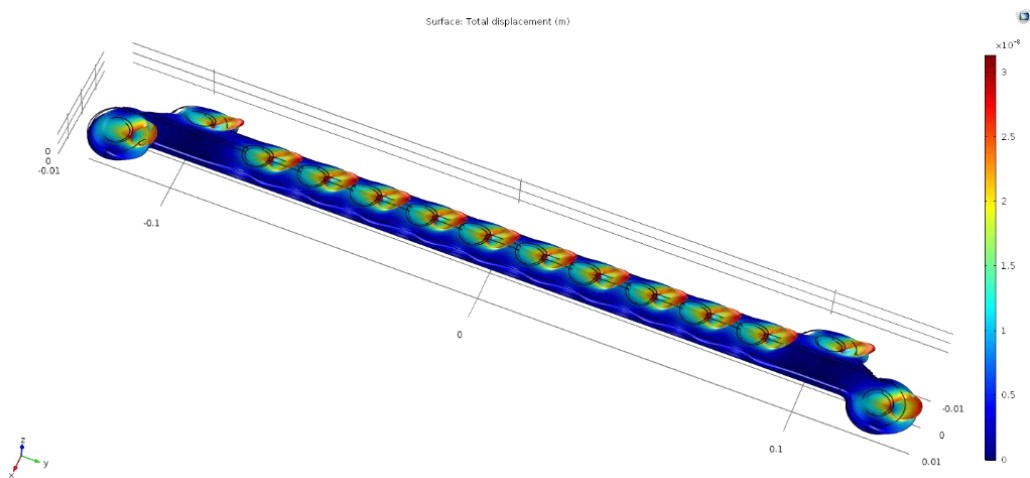

**Figure 18.** Total displacement of newly developed plate bent for 30°.

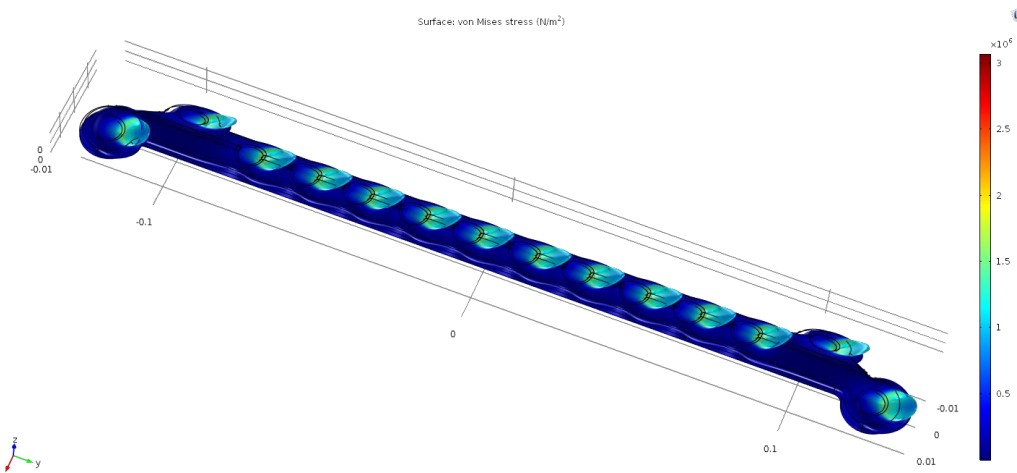

**Figure 19.** Von Mises stress of newly developed plate bent for 30°.

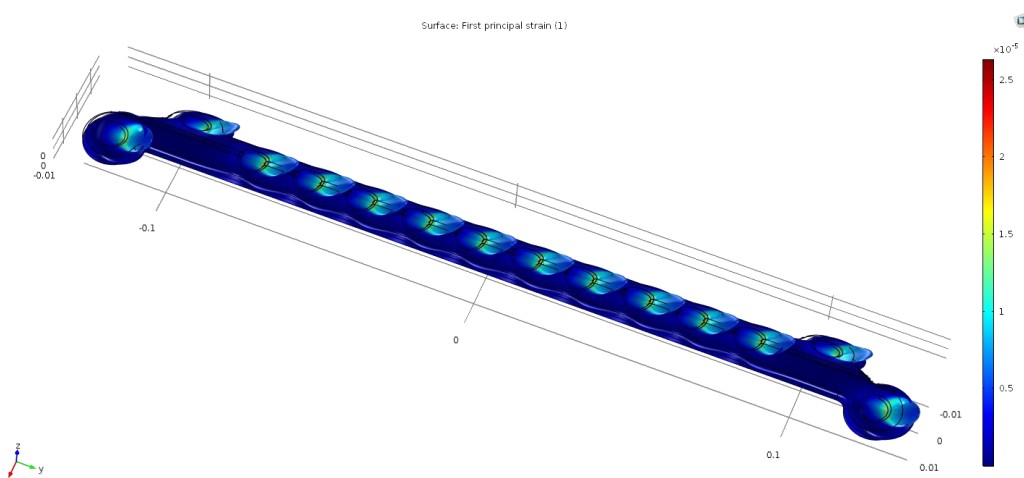

**Figure 20.** First principal strain of newly developed plate bent for 30°.

Bent, 40°

Figures 21–23 illustrate the simulation result of the newly developed bone plate, bent at 40°. Similar with the 30° bent plate, the maximum displacement of the 40° bent plate is only 34%, and the average displacement is 134% of the traditional flat plate. Similarly, the maximum von Mises stress and first principal strain are only 21% and 15%, respectively, but the average value is 122% and 123% of the traditional flat plate, respectively. When compared with the newly developed flat plate, this plate has a maximum displacement of only 87%, maximum von Mises stress of only 51%, and maximum first principal strain of only 34%. The average displacement is 137%, while the average von Mises stress and average first principal strain are 122% and 129%, respectively, in contrast to the newly developed flat plate. Although there is improvement in the maximum value when compared with both the traditional flat plate and the newly developed flat plate, the average value is still greater than those two, which will make the bone plate not an ideal choice for clinical use.

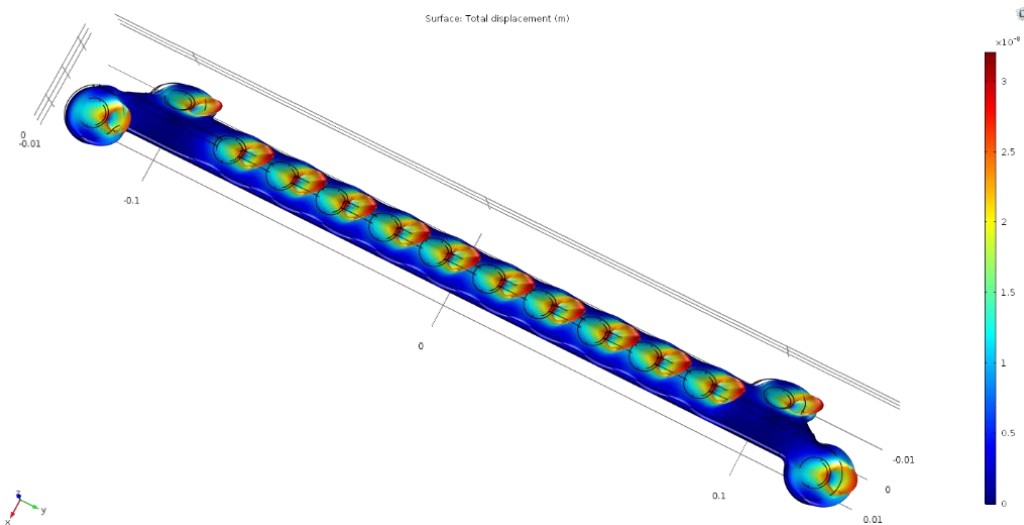

**Figure 21.** Total displacement of newly developed plate bent for 40°.

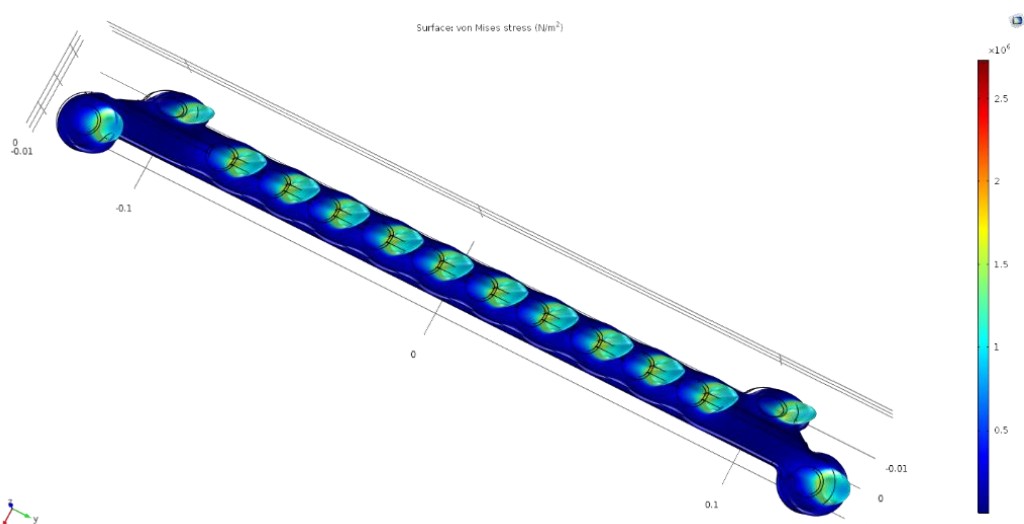

**Figure 22.** Von Mises stress of newly developed plate bent for 40°.

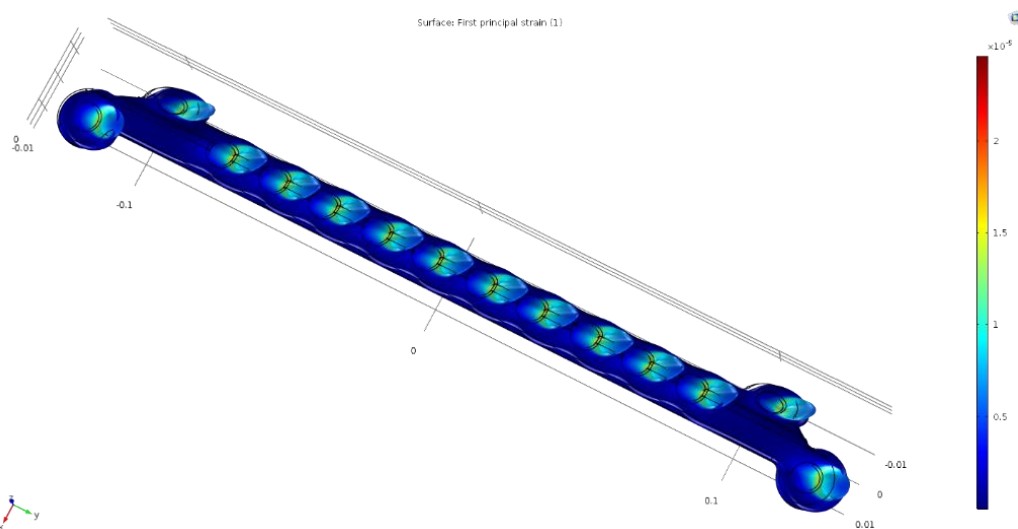

**Figure 23.** First principal strain of newly developed plate bent for 40°.

Bent, 50°

Finally, the plate with a 50° bending angle has a result that can be seen in Figures 24–26. Generally, this plate has the highest displacement compared with the developed plate—both flat and bent—and the highest von Mises stress and first principal strain compared with all bone plates analyzed—including the traditional flat plate. This shows that this plate will not help much in a clinical situation and it might even worsen the patient's condition.

*3.3. Result Summary*

Overall, the simulation result is summarized in Tables 2–4, from which data are extracted to develop the graph illustrated in Figures 27–29.

3.3.1. Total Displacement (m)

Table 2 shows the result of the total displacement analysis, which is also illustrated in the graph in Figure 27. It is clear that the plate with the greatest maximum displacement is the traditional plate, and the maximum displacement of the newly developed plate is increased when the bending angle is also increased. All simulated plates have zero as the minimum displacement, while the average displacement fluctuated slightly with the difference in the design and bending angle.

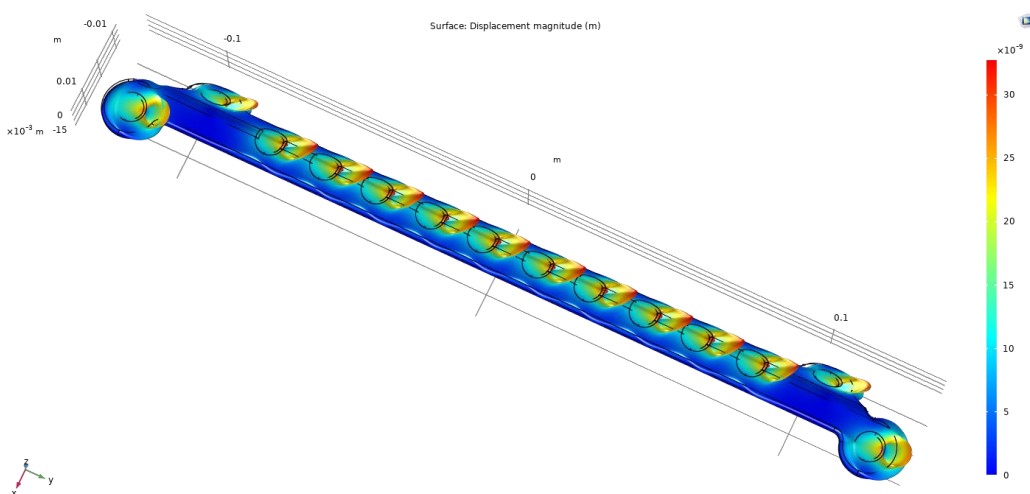

**Figure 24.** Total displacement of newly developed plate bent for 50°.

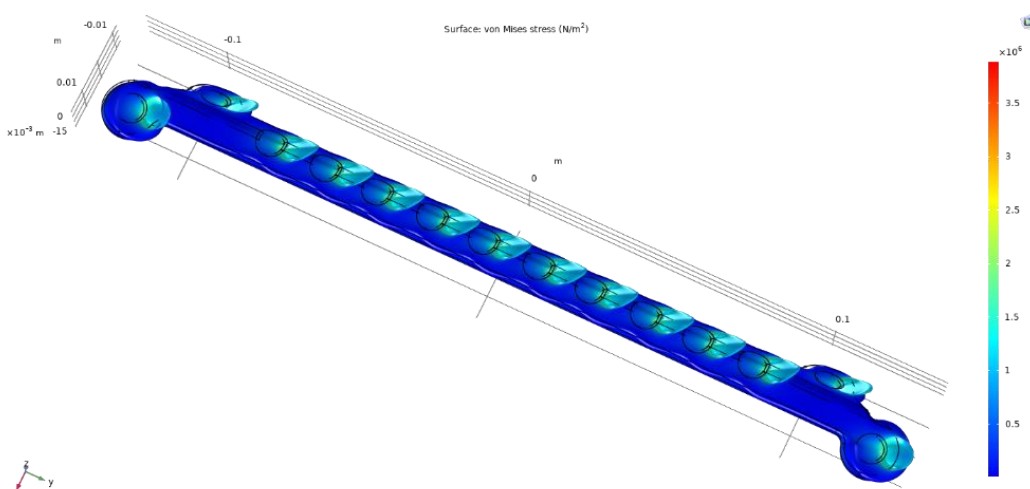

**Figure 25.** Von Mises stress of newly developed plate bent for 50°.

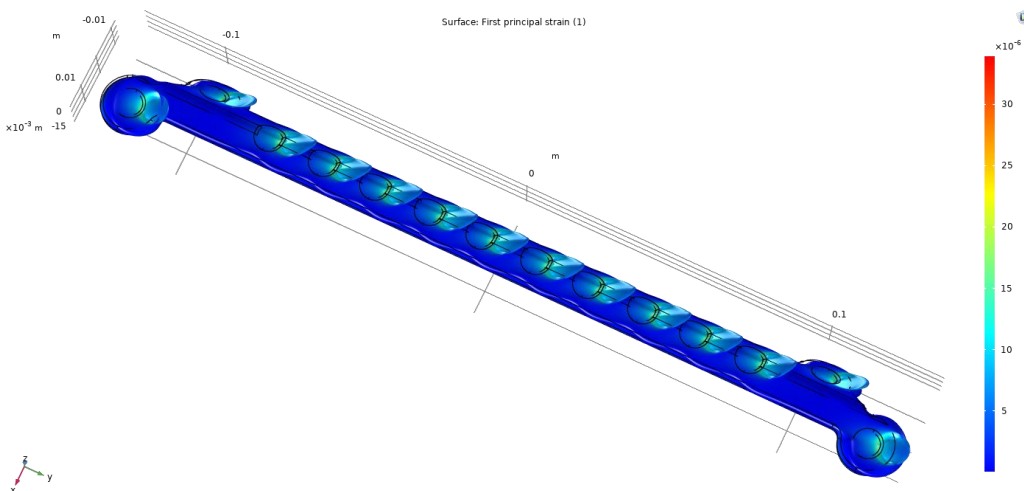

**Figure 26.** First principal strain of newly developed plate bent for 50°.

**Table 2.** Total displacement (m) of each plate.

| No. | Name | Minimum | Maximum | Average |
|---|---|---|---|---|
| 1 | Traditional | 0 | $48.8 \times 10^{-9}$ | $3.63 \times 10^{-9}$ |
| 2 | Design—0° | 0 | $29.6 \times 10^{-9}$ | $4.53 \times 10^{-9}$ |
| 3 | Design—10° | 0 | $29.6 \times 10^{-9}$ | $4.61 \times 10^{-9}$ |
| 4 | Design—20° | 0 | $30.5 \times 10^{-9}$ | $4.69 \times 10^{-9}$ |
| 5 | Design—30° | 0 | $31.4 \times 10^{-9}$ | $4.77 \times 10^{-9}$ |
| 6 | Design—40° | 0 | $32.1 \times 10^{-9}$ | $4.86 \times 10^{-9}$ |
| 7 | Design—50° | 0 | $32.7 \times 10^{-9}$ | $4.95 \times 10^{-9}$ |

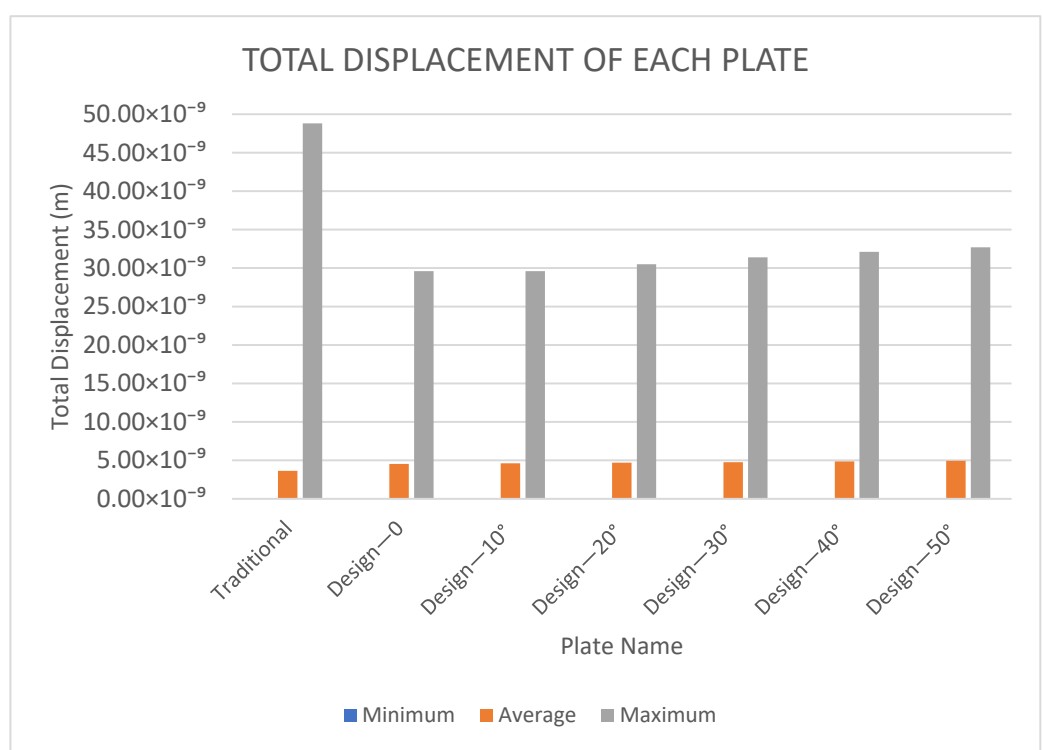

**Figure 27.** Total displacement (m) of each plate.

### 3.3.2. Von Mises Stress ($N/m^2$)

The von Mises stress is summarized in both tabular form (Table 3) and in a bar chart (Figure 28). As illustrated in Figure 28, the newly developed plate has the least maximum von Mises stress compared with all of the simulated bone plates, and the newly developed plate bent at 50° has the highest value of von Mises stress both on average as well as at a maximum, while also having the lowest minimum value.

### 3.3.3. First Principal Strain (1)

Table 4 summarizes the first principal strain of all plates, and the data are illustrated in a bar chart in Figure 29. Similar to the von Mises stress, the lowest maximum first principal strain is on the newly developed plate. The first principal strain from the FEA simulation result of the newly developed plate bent at 50° is also very much alike the von Mises stress result.

**Table 3.** Von Mises stress (N/m$^2$) of each plate.

| No. | Name | Minimum | Maximum | Average |
|---|---|---|---|---|
| 1 | Traditional | $6.02 \times 10^{-1}$ | $3.57 \times 10^6$ | $0.227 \times 10^6$ |
| 2 | Design—0° | $7.04 \times 10$ | $2.07 \times 10^6$ | $0.269 \times 10^6$ |
| 3 | Design—10° | $6.56 \times 10$ | $2.88 \times 10^6$ | $0.271 \times 10^6$ |
| 4 | Design—20° | $3.95 \times 10$ | $3.84 \times 10^6$ | $0.273 \times 10^6$ |
| 5 | Design—30° | $6.11 \times 10$ | $3.09 \times 10^6$ | $0.276 \times 10^6$ |
| 6 | Design—40° | $4.94 \times 10$ | $2.81 \times 10^6$ | $0.278 \times 10^6$ |
| 7 | Design—50° | $2.15 \times 10$ | $4.10 \times 10^6$ | $0.280 \times 10^6$ |

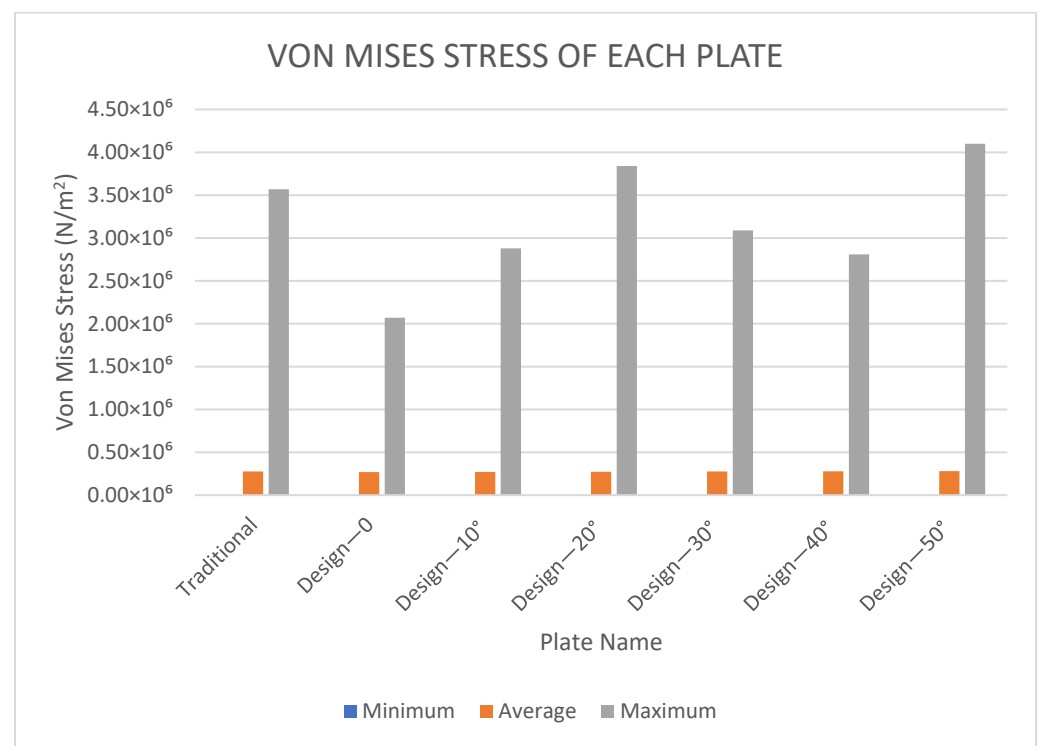

**Figure 28.** Von Mises stress (N/m$^2$) of each plate.

**Table 4.** First principal strain (1) of each plate.

| No. | Name | Minimum | Maximum | Average |
|---|---|---|---|---|
| 1 | Traditional | $1.99 \times 10^{-12}$ | $29.3 \times 10^{-6}$ | $1.48 \times 10^{-6}$ |
| 2 | Design—0° | $3.60 \times 10^{-10}$ | $16.4 \times 10^{-6}$ | $1.75 \times 10^{-6}$ |
| 3 | Design—10° | $3.40 \times 10^{-10}$ | $23.6 \times 10^{-6}$ | $1.77 \times 10^{-6}$ |
| 4 | Design—20° | $1.35 \times 10^{-10}$ | $33.3 \times 10^{-6}$ | $1.79 \times 10^{-6}$ |
| 5 | Design—30° | $3.91 \times 10^{-10}$ | $26.3 \times 10^{-6}$ | $1.81 \times 10^{-6}$ |
| 6 | Design—40° | $2.83 \times 10^{-10}$ | $24.9 \times 10^{-6}$ | $1.83 \times 10^{-6}$ |
| 7 | Design—50° | $1.41 \times 10^{-10}$ | $35.2 \times 10^{-6}$ | $1.85 \times 10^{-6}$ |

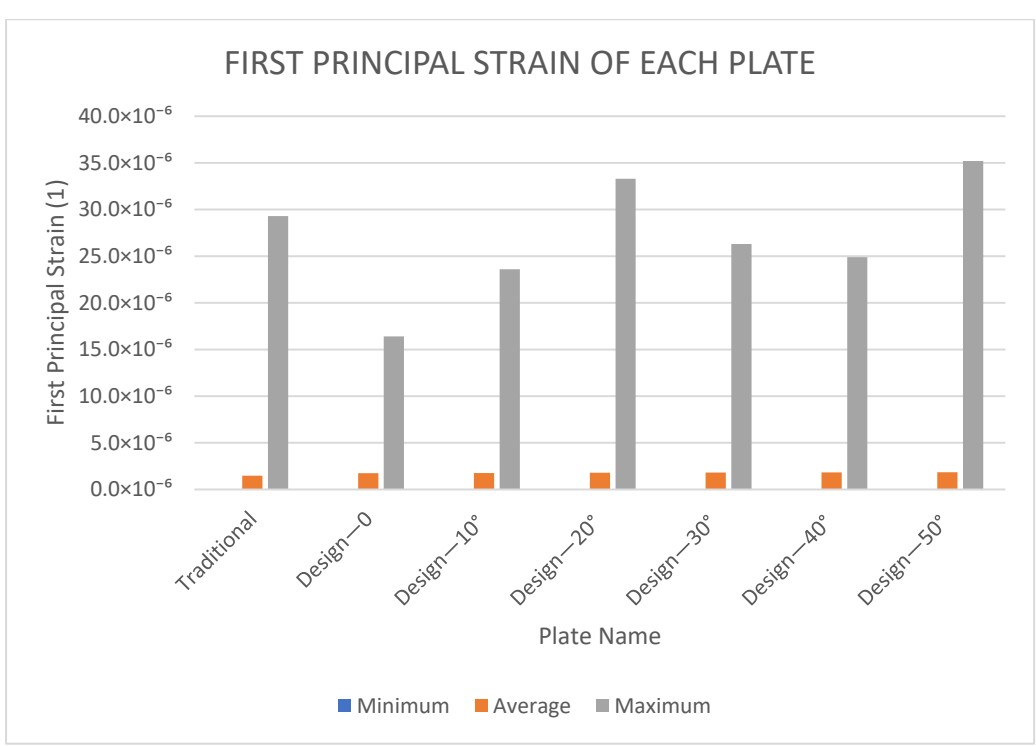

**Figure 29.** First principal strain (1) of each plate.

Overall, all of the bone plates were tested using COMSOL Multiphysics and each has different degrees of displacement, von Mises stress, and first principal strain. Although every plate experienced a certain degree of deformation, it is observed that no plate fractured during the simulation. The main criteria that should be possessed by a bone plate exists in all plates here, meaning that all of these bone plates could actually be used clinically. All bone plates experience damages, mainly on the screw holes towards the end of the bone plate, as the load only applied to that location with the proximal/distal direction is the greatest. This is also the reason that the deformation on the screw holes looks slanted to one side, towards the proximal/distal end. Nevertheless, there is another consideration that makes a certain bone plate superior to the other, such as the form and the degree of the deformation.

After comparing the flat plates, the developed flat plate is considered to be better in terms of total displacement, von Mises stress, and first principal strain. The result explained above shows that the developed plate has less overall deformation, while also being stronger and more stable than the traditional flat rectangular plate. Furthermore, the overall result shows that the bone plate bent at 10° in the middle part shows a superior quality, mainly in the maximum value of all aspects, while the average value can still be improved. It could be concluded that the bone plate with a 10° bending angle would be the best candidate for clinical use, in harmony with the essential properties of a bone plate that should not be easily broken, in order to ensure a speedy recovery. The highest stress acting on the plate is only 2.88 MPa, averaging at 0.271 MPa, which is less than the stress limit for Ti-6Al-4V before its plastic deformation. The maximum displacement is also only 29.6 nm and average of 4.61 nm, with a maximum first principal strain of $23.6 \times 10^{-6}$. Therefore, the bone plate implant bent at 10° in the middle would be a better solution for patient use, particularly in the left femoral bone, instead of the traditional flat implant. The FEA result, however, is still not fully independent and a physical mechanical test should also be conducted to ensure the reliability of this implant design.

## 4. Conclusions

A new bone plate implant is designed to be stronger and more stable for patient use, in comparison with the traditional rectangular bone plate. The new design is also bent in the middle so it can be observed if it is more stable than the regular flat plate. The result shows that the new design is better in terms of strength and balance. The bent developed plate with a 10° bending angle is shown to be the best compared with all bent bone plates, with an average total displacement of 4.61 nm, average von Mises stress of 0.271 MPa, and average first principal strain of $1.77 \times 10^{-6}$. In conclusion, the novel design bone plate with a 10° bending angle is the most suitable bone plate for patient use compared with the traditional flat rectangular plate.

**Author Contributions:** Conceptualization, methodology, analysis, writing, J.K.; conceptualization, methodology, supervision, S.-Y.L. and W.-T.W. All authors have read and agreed to the published version of the manuscript.

**Funding:** The authors acknowledge the National Formosa University for their support in this research.

**Institutional Review Board Statement:** Not applicable.

**Informed Consent Statement:** Not applicable.

**Data Availability Statement:** Correspondence and requests for materials should be addressed to Shen-Yung Lin.

**Conflicts of Interest:** The authors declare no conflict of interest.

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
