# Peer review of "Development and Finite Element Analysis of a Novel Bent Bone Plate"

_applsci, doi:10.3390/app122110900_

Round 1
Reviewer 1 Report
Title: Development and Finite Element Analysis for a Novel Bent Bone Plate
Author: Joyceline Kurniawan, Shen-Yung Lin and Wen-Teng Wang
After reviewing the manuscript, there are some mistakes. The detail comments are as follows:
1. In Table 2, the unit of Yield Strength and Tensile Strength should be MPa instead of GPa, please revise. In addition, are the references [3] and [16] or the references [17] and [18] used for the mechanical Properties of Titanium Alloy 6Al-4V presented in Table 2? Please unify.
2. In Table 3 and Figure 27, please give the calculation formula of the average displacement.
3. In reference [17], plte should be plate.
4. The quality of the figures are poor, and need to be redrawn.
Reviewer 2 Report
This paper shows high sigificance of content and high quality of presentation.
I think
I think this paper can be published.
Author Response
No comments should be revised.
Reviewer 3 Report
1. In this paper, the authors develop a new bone plate implant design with middle bending. The explanation of traditional flat plate features is not enough, and more previous research on flat plate or bending plate should be given in the introduction section.
2. The authors have tested plates using a FEA software, so the details of COMSOL Multiphysics need to be given, such as the version number, company name and so on.
3. In this manuscript, the bending angle is used in different plates, and what is the methodology based on and are there any relevant studies?
4. In the “result summary” part, it said that “It would be essential for a bone plate to not be easily broken, especially in the fixation spot, in order to ensure a successful recovery.” Please clarify the more reliable basis, like other previous studies.
5. The manuscript may need to add a discussion part, and this part will make the results more useful for other researchers.
Round 2
Reviewer 1 Report
The manuscript has been revised according to reviewer's comments, and can be accepted in this journal.